# Effect of Cerium on the Microstructure and Inclusion Evolution of C-Mn Cryogenic Vessel Steels

**DOI:** 10.3390/ma14185262

**Published:** 2021-09-13

**Authors:** Liping Wu, Jianguo Zhi, Jiangshan Zhang, Bo Zhao, Qing Liu

**Affiliations:** 1State Key Laboratory of Advanced Metallurgy, University of Science and Technology Beijing, Beijing 100083, China; lipingw2021@126.com (L.W.); zjsustb@163.com (J.Z.); 2Centre of Technology Baotou Iron and Steel Co., Baotou 014010, China; jianguoz2021@126.com; 3Inner Mongolia Enterprise Key Laboratory of Rare Earth Steel Products Research and Development, Baotou 014010, China; 4Institute of Metallurgical Technology, Iron and Steel Research Institute Beijing, Beijing 100083, China; zhao_working@163.com

**Keywords:** cryogenic vessel steels, cerium, microstructure, inclusions

## Abstract

The effects of Cerium (Ce) were studied on the casting slab quality, microstructure, and inclusion evolution of cryogenic vessel steel. An optical metallographic microscope, scanning electron microscope, energy dispersive spectrometer, and Thermo-calc thermodynamic software were used for characterization and analysis. The results indicated that the central segregation was significantly improved after adding Ce and reached the lowest level when the content of Ce was 0.0009 wt.%. Meanwhile, the presence of Ce reduces the size of ferrite and improves pearlite morphology. Ce also enables the modification of Al_2_O_3_ and MnS + Ti_4_C_2_S_2_ inclusions into ellipsoid CeAlO_3_ and spherical Ce_2_O_2_S + Ti_4_C_2_S_2_ composite inclusions, respectively, which are easier to remove. The formed Ce_2_O_2_S inclusions are fine and can work as heterogeneous nucleation points to refine the microstructure of steel.

## 1. Introduction

Cryogenic vessel steel is an essential material for producing storage tanks and is widely used to store and transport liquefied gases. As increasing clean energy sources are developed and applied, cryogenic vessel storage equipment has been finding more applications [1,2,3,4,5]. Cryogenic vessel storage equipment requires stringent demands for the alloy composition, casting slab quality, and mechanical properties of the cryogenic vessel steel.

Importantly, inclusions affect the cleanliness, control, and quality of steel. During the solidification-rolling process, the morphology of inclusions is changed. Rare earth (RE) elements present a remarkable effect on modifying inclusions and in facilitating clean steel production, given their unique electronic layer structures. Earlier investigations have reported that different RE elements could purify molten steel, modify inclusion morphology, increase cryogenic toughness, and improve corrosion resistance in different steel series [6,7,8,9,10]. When RE is added to extra-low sulfur (S < 0.003%) niobium-titanium micro-alloy steel, the deoxidation and inclusion removal rates can be increased, therefore the molten steel can be more effectively purified [11]. After adding La+Ce compounds to ship plate steel, the inclusions mainly exist in the form of spheres and approximate spheres, while the sharp-angled S-Mn-Ca phase and S-O-Mn-Ca-Al phase inclusions are reduced and the hot-rolled steel strip band structure is lightened [12]. The type of inclusions in drill steel without Ce additions are MgAl_2_O_4_ and (Ca, Mn)S. As the Ce content in drill steel reaches 0.0078% (mass fraction), the types of inclusions change to Ce-O and Ce-S and the size of the inclusions in the drill steel decrease significantly [13]. Inclusions affect the toughness, corrosion resistance, and fatigue properties of steel, and reducing their size weakens the impact of non-metallic inclusions on the properties of steel. The RE elements La, Ce, and Nd play deoxidation and desulfurization roles in molten steel, making the inclusions fine and diffuse in distribution [14]. Gao et al. found that after Ce treatment, irregular Al_2_O_3_ inclusions with a size of 10−15 μm are wrapped by rare earth and then gradually modified into spheroidal CeAlO_3_, Ce_2_O_3_, and Ce_2_O_2_S inclusions with a size of ≈5 μm, distributed into interstitial free slabs [15].

It has been proposed in the literature that RE influences the microstructure of a casting slab. For example, with the addition of La into nonoriented electrical steel, the equiaxed crystal ratio increased and the columnar crystals were suppressed and refined [16]. The addition of La+Ce changed the solidification microstructure distribution of thin slabs produced by compact strip production and refined the solidification structure of 430 ferritic stainless steel, thereby improving the comprehensive properties of the material [17,18]. The addition of La had a significant effect on the distribution of arsenic by the formation of arsenic inclusions, and various RE have been widely used in the steelmaking process to improve steel quality [19].

However, there is still no systematic explanation regarding the mechanisms by which RE improves the quality of casting slabs, changes the structure morphology, and modifies the inclusions. Moreover, casting slab quality is the most important index to evaluate the finished product. The quality of a casting slab should be analyzed from a macro quality perspective, microstructure characteristics, and the evolution law of the inclusions. In this study, the characteristics of inclusions and the microstructure of C-Mn cryogenic vessel steels were comparatively studied without Ce and with the addition of different Ce content levels.

## 2. Experimental

The steel samples were taken from a Inner Mongolia Baotou Iron and Steel Group Company production site, located in Baotou, China. The production process uses a 240 t top and bottom combined blow basic oxygen furnace (BOF), dephosphorizing with the bottom blow, and performing automatic steelmaking with sublance 240 t ladle furnace (LF) refining, and 240 t Ruhrstahl–Heraeus (RH) refining treatment for a minimum of 25 min. The RE ferrocerium alloy (where the mass fraction of Ce is 30% and the total mass fraction of impurity elements including sulfur, oxygen, and phosphorus is lower than 0.01%, the rest being Fe) were added into the ladle at the end of vacuum treatment, and then re-pressed for 5 min. After re-pressing and soft blowing for greater than 8 min to homogenize the composition, it was ensured that the inclusion as fully floated and improved the cleanliness of the molten steel. During the continuous casting process of the slab, protective casting measures prevented an increase in nitrogen and oxygen. The casting speed was 1.2 m·min^−1^, and the casting slab section dimensions were 230 × 1550~2030 mm, using a 1650 mm double stand slab caster, respectively. The chemical composition of the tundish sample was analyzed by direct reading spectrum analyzer (ARL-4460, Thermo Fisher Scientific, Geneva, Switzerland), and the actual content of Ce in the steel was detected by chemical analysis (ICP-MS, Plasma Mass Spectrometer, Waltham, Massachusetts, America Perkin Elmer). Table 1 shows the chemical compositions of the experimental steel. The hot-rolled steel strip was produced on a 2250 mm rolling mill using controlled rolling and controlled cooling technology. The discharging temperature was 1190 °C, the finish rolling temperature was 840 °C, the coiling temperature was 600 °C, and the thickness of the finished product was 12 mm.

The casting slab selected had a size of 300 mm × 1500 mm × 230 mm. After hot acid corrosion, casting slab quality was determined according to the standard diagrams for macrostructure and defects in a continuous casting slab (using YB/T 4003-2016, which is a kind of steel industry criterion in China, equal to ASTM E381-2017). In this study, we took one piece of steel at 3/4 of the width of the casting slab, with dimensional specifications of 100 mm × 100 mm × 230 mm, and we also chose three metallographic samples at 1/4 distances from the inner arc, slab center, and 1/4 distance from the outer arc, with dimensional specifications of 10 mm × 10 mm × 10 mm, following the specific sampling plan shown in Figure 1.

The sampling position of the hot-rolled finished plate corresponds to the casting slab, which is located at 3/4 of the width direction of the steel plate, and the sample size was 12 mm × 10 mm × 10 mm. The samples were mechanically ground (from #400~#1000) and then polished. Then samples were etched using 4% nital. Optical microscope (OM, Axio Observer D1 m, Jena, Germany) analysis was used to observe the microstructures. Inclusion distribution and morphology were determined using a scanning electron microscope (SEM, LEO EVO, Germany) and a field emission scanning electron microscope (FE-SEM, SIGMAN300, Carl Zeiss Company, Oberkochen, Germany). The compositions of inclusions were identified with an energy dispersive spectrometer (EDS, GENESIS, America).

In order to more intuitively show the effect of RE elements on the denaturation of other inclusions, Thermo-calc thermodynamic software was used to calculate the variation law of inclusions in C-Mn cryogenic vessel steel with the addition of Ce. Combined with the experimental results, the evolution law of various inclusions was summarized, and a dominant zone diagram of Ce inclusions was obtained.

## 3. Results and Discussion

### 3.1. Effect of Ce on the Casting Slab Quality

Casting slab results, with and without Ce, are shown in Figure 2. There are apparent differences in the observation results of the samples. It can be seen that the central segregation of the casting slabs with the Ce addition had a tendency to be ameliorated. Especially after adding 0.0009 wt.% Ce, the shape of the central segregation line almost disappeared, and the internal quality of the casting slab was excellent.

There are usually three types of central segregation, A represents the most serious, B is the second, and C is the lightest. Each type is divided into three levels, the smaller the number, the lighter the central segregation. The center porosity is the incompact structure formed by the final crystal shrinkage of the casting slab. The degree of porosity was judged by the size of the gap, and the casting slab quality was evaluated. Table 2 quantitatively shows the judgment results for the casting slabs. It can be seen that the central segregation of the casting slab was greatly improved after Ce treatment. The results reveal that the RE Ce plays a vitally important role in the casting slab.

Central segregation determines the quality of a casting slab. When high temperature stress exceeds the maximum stress that the casting slab can withstand, cracks will first form near the central segregation zone and then extend outward gradually. Cracks indirectly affect the uniformity of composition, the structure of hot-rolled finished product and the judgment level of the band structure. During the period of solidification, Ce reduces the residual sulfur and oxygen content in molten steel. High melting point RE sulfides and oxysulfides are formed which can hinder the segregation of the alloying elements, suppressing the accumulation of sulfides at the grain boundaries. Many scholars [20,21,22] have indicated that the formation of Ce-O-S inclusions increases the heterogeneous nuclei particles, improves the solidification structure, and reduces the center segregation of the casting slab.

Under the same casting conditions, part of the Ce is dissolved within the molten steel, reducing the segregation of C and Mn, while the other part of the Ce forms the RE inclusions. The melting point of the inclusions formed by the RE elements is also higher than that of the general metal compounds. The RE inclusions precipitate and become dispersed within the molten steel during the cooling process. The RE inclusions become nucleated particles. The outer layer will gradually become encased by the other metal inclusions, thereby optimizing the solidification structure of the molten steel and improving the quality of the casting slab. Previous researchers have indicated that adding the appropriate amounts of RE elements into weathering steel significantly improved central segregation [23].

The internal quality and macrostructure of the casting slab are closely related to the microstructure of the product. Figure 3 shows the microstructure with and without and Ce at different positions of the casting slab thickness. It can be seen that as the content of Ce increases, the proportion of pearlite increases and the proportion of ferrite decreases, however the microstructure tends to be refined. According to the reports, after the addition of mixed La and Ce elements into 0.27C-1Cr steel, the original austenite grains were refined, the grain growth rate was slowed, the nucleation rate of the phase transformation products increased, Ac_1_ was unchanged, Ac_3_ increased, the Ms point decreased, the pre-eutectoid ferrite or bainite precipitation period shortened, and the pearlite and bainite transformation completion time were prolonged [24]. 

These results mean that the addition of Ce can change the structural characteristics of a casting slab. Due to a unique electronic layer structure, the edge-to-edge matching model [25] was used to calculate it. The results show that containg-Ce oxide is potential nucleant for the heterogeneous nucleation of both the δ-Fe and γ-Fe primary phases during the solidification of steels, and it promotes grain refining. Also, Ce promotes the formation of random hexagonal close packing structures during the nuclei formation process, and thus helps to reduce nucleation-free energy [26]. As shown in Figure 3, the grains will be coarsened with the increase of Ce. However, when Ce addition is 0.0009 wt.%, the pearlite and the ferrite are finer and more uniform. This is primarily due to the addition of Ce in appropriate amounts in order to form a fine RE composite inclusion. This promotes the nucleation of acicular ferrite in the austenite body, divides the austenite, and refines the grain. Also, under the same casting conditions, the Ce may be responsible for increasing the transformation temperature of austenite to pearlite. It can be concluded that the addition of the optimum content of Ce played a vital role in the size and shape of the microstructure of the casting slab. As discussed in previous work, the distribution of RE on the grain boundaries indicates a refinement to pearlite layer spacing and ferrite microstructure [27,28].

Figure 4 shows the morphology of the inclusions without and with the addition of Ce. From Figure 4a, on the metallographic samples, islands it can be seen that MnS inclusions formed near the shrinkage holes and occasionally were intermittently connected [20]. From Figure 4b, on the metallographic samples, the shape of Al_2_O_3_ inclusions was irregular and blocky. These inclusions deteriorate matrix continuity and are origins for the formation of cracks. For the optical microstructure of the samples, after the addition of Ce, the casting slab inclusions were not prismatic or chain-shaped but ellipsoidal. When the steel is hotly processed and cooled, due to the RE inclusions, the thermal expansion coefficient and density were close to that of molten steel [29,30], and it could thus avoid the stress around the inclusions, significantly reducing harm to the material. Figure 4 presents the inclusions in the (c), (d), and (e) steel samples after adding the Ce alloy to molten steel. The morphology of these inclusions was gradually spheroidized compared to those without RE, which greatly reduced the damage to the steel matrix.

To determine the composition of the complex inclusion mentioned above clearly through further investigation, the inclusions were analyzed using SEM-EDS to determine their chemical composition, and the results are shown in Figure 5. The Al_2_O_3_ inclusions in the sample without Ce added were sharp-angled, while the MnS+Ti-C-S inclusions were irregular, as shown in Figure 5a,b, respectively. The addition of RE Ce modifies the Al_2_O_3_ inclusion morphology, forming CeAlO_3_, which are nearly spherical inclusions. Simultaneously, the affinity of Ce to sulfur is stronger than it is to manganese [31], and thus CeAlO_3_ or Ce_2_O_2_S spherical inclusions were formed instead of MnS + Ti-C-S inclusions. When an appropriate amount of Ce was added to the molten steel, the Ce in the molten steel modified the Al_2_O_3_ and series MnS inclusions, forming Ce complex inclusions, making their appearance approximately spherical. Some studies have indicated that the inclusions with an aspect ratio less than or equal to √2 would be treated as circular, and the other inclusions will be treated as elliptical [32]. The typical morphology and composition of inclusions with Ce additions are shown in Figure 5c–e. The irregular Al_2_O_3_ inclusions or MnS + Ti-C-S inclusions were wrapped by Ce, and thus gradually transformed into spheroidal Al-O-Ce-Mg-Ca, Al-O-Ce-C-Ca or Al-O-Ce-C-Ca inclusions. With the content of Ce increasing in molten steel, the inclusions were gradually modified to RE inclusions, and the size of the inclusions decreased. But when the Ce was over 0.0009 wt.%, the size of the containing-Ce inclusions shows an increased and irregular trend. It could be concluded that the appropriate addition of Ce modified the inclusions.

### 3.2. Effect of Ce on Microstructure and Inclusions in Steel Strips

Figure 6 shows the microstructures of hot-rolled steel strips with and without different Ce levels. It can be seen that the proportion of pearlite is small and the size of the structure is uneven in Figure 6a. As shown in Figure 6b–d, the proportion of pearlite is greater. The proportion of pearlite and ferrite were measured quantitatively by the Photoshop CC (Creative Cloud) image analysis tool, and the results are as shown in Figure 7. It can be found that with the increase of the Ce element, the content of pearlite shows an increasing trend, and the content of ferrite shows a decreasing trend. Meanwhile, when the Ce was 0.0009 wt.%, the presence of Ce reduced the size of ferrite, and the microstructure was uniform and fine. It can be concluded that the microstructure in the casting slab is directly inherited after hot rolling, such that more Ce is added, the better and coarser the microstructure. One possible reason is that in the C-Mn cryogenic vessel steels, Ce promotes the diffusion of C, which is conducive to the formation of pearlite, and the formation of fine Ce inclusions improves the nucleation of pearlite and finer ferrite. Zang et al. reported that the addition of Y element causes a large ferrite structure to appear in 20Cr13 martensitic stainless steel [33].

The above results show that Ce can be a modified inclusion in the casting slab. Figure 8 shows the typical nonmetallic inclusions after the corresponding casting slab is rolled into the finished steel strip. Figure 8a,b show the without adding Ce inclusions, the shape is irregular and rectangular; these kind of inclusions are more harmful to the steel matrix and also a sensitive source of crack propagation. Figure 8c–e show ellipsoidal RE non-metallic inclusions observed in the microstructure after adding different contents of Ce, which reduces the possibility of crack formation and propagation [34]. The size of the inclusions decreases and tends to spheroidize, and when Ce is 0.0009 wt. %, the shape of inclusions is spherical and their size is the smallest. These data indicate that the addition of an appropriate quantity of Ce played an imporptant role in the size distribution and shape deformation of the inclusions.

Combining the casting slab microstructure analysis in Figure 4 and Figure 5, it was concluded that Ce additions modified the grain size of the studied steels containing them. Meanwhile, after adding Ce, large particles of inclusions floated up, so a typical finer spherical inclusion morphology was chosen for the surface scan analysis as shown in Figure 9, which demonstrates that the inclusions should be Ce_2_O_2_S. These finer RE inclusions act as heterogeneous nucleation points [35], reducing the distribution of harmful elements in the steel matrix.

### 3.3. Ce Inclusions Formed Thermodynamic Mechanisms

To illustrate the evolution mechanism of the inclusions, Thermo-Calc software was used to calculate the formation results of S-containing inclusions in C-Mn cryogenic vessel steel with and without the addition of RE Ce. As shown in Figure 10 and Figure 11, it was found that after adding Ce, the contents of Ti_4_C_2_S_2_ and MnS#2 type inclusions were significantly reduced, forming high melting point Ce_2_O_2_S and Ce_2_S_3_ inclusions, while MnS#2 is a composite inclusion phase dominated by MnS. One possible reason is that the RE data of the thermodynamic software is not comprehensive, which leads to certain limitations in the calculation results.

To further explore the influences of different Ce contents on the evolution of inclusions [36], the Gibbs free energy of the reactions under different Ce contents (0–100 ppm) was calculated. The values of other elements were selected from Table 1 (No.II). The results are shown in Figure 12.

From Figure 12, the content of the RE elements has little effect on the precipitation order of the inclusions. After the initial addition of a small amount of Ce, the Gibbs free energy of each reaction was significantly reduced. However, increasing by the amount of Ce added, the change in the Gibbs free energy gradually became smaller. According to the relative size of the Gibbs free energy, it can be concluded that CeAlO_3_ is the most stable, followed by Ce_2_O_3_, then Ce_2_O_2_S.

The formula ΔG=−RTlnK and previous research data [37,38,39] were used to calculate the conditions for the formation of inclusions, and the results are shown in Table 3. The phase diagram calculation results of Ce-O-S at a temperature of 1873K are shown in Figure 13.

Combining Table 3 and Figure 13, the conclusions are as follows:

(1) Formation conditions of RE oxides:when a_[O]_ > 0.232, a_[S]_/a_[O]_^2^ < 98.329 the inclusions generated are CeO_2_, when a_[O]_ < 0.232, a_[S]_/a_[O]_ < 12.67, the inclusions generated are Ce_2_O_3_.

(2) Formation conditions of RE oxysulfides:when a_[O]_ < 0.232, 12.67 < a_[S]_/a_[O]_ < 262.433 or a_[S]_ < 0.5857, 98.3284 < a_[S]_/a_[O]_^2^ < 117580, the inclusions generated are Ce_2_O_2_S.

(3) Formation conditions of RE sulfides:when a_[S]_ < 0.5857, a_[S]_/a_[O]_^2^ > 117580, the inclusions generated are CeS, and when a_[S]_ > 0.5857, a_[S]_/a_[O]_ > 262.433, the inclusions generated are Ce_2_S_3_.

### 3.4. Inclusion Formation Mechanisms with the Addition of Ce

RE elements have a strong mutual chemical affinity to [O] and [S] [40]. Both elements begin to aggregate and form high melting point oxides, oxysulfides and sulfides, thus having the ability to remove the deleterious types of inclusions that are formed [13,32,41]. The combination of Ce with [O] and [S] in steel has lower Gibbs free energy. It was easy to change the inclusion morphology from chains and strips to generate CeAlO_3_, Ce_2_O_2_S, and Ce_2_O_3_ spherical inclusions [42]. The RE inclusions CeAlO_3_ and Ce_2_O_2_S in molten steel were stable at 1873K, decreasing the mass fraction of dissolved oxygen in the molten steel, while Ce and S formed complex RE sulfides, resulting in a decrease in CeAlO_3_ and increase in Ce_2_O_2_S [43,44]. According to the discussion in Section 3.2 and Section 3.3, Ce_2_O_2_S inclusion is especially beneficial for heterogeneous nuclei. The evolution process of RE inclusions during solidification is shown in Figure 14.

Firstly, when the melting temperature is stable at 1873K, it is easy to form Al_2_O_3_ and MnS+Ti_4_C_2_S_2_ inclusions in molten steel. As shown in Figure 13 (Stage I), the inclusions are primarily irregularly shaped and deteriorate the mechanical properties of steel [45]. The reaction equation is as follows:[S] + [Mn] = MnS_(S)_(1)
4[Ti] + 2[C] + 2[S] = Ti_4_C_2_S_2(S)_
(2)
2[Al] + 3[O] = Al_2_O_3(S)_
(3)

Secondly, when RE-Ce alloy are added to the molten steel, large-sized irregular inclusions are also transformed into small-sized spherical or ellipsoidal CeAlO_3_ and Ce_2_O_2_S inclusions via Ce. The reaction equation [37,38] is as follows:[Ce] + 3[O] + [Al] = CeAlO_3(S)_
(4)
2[Ce] + 2[O] + [S] = Ce_2_O_2_S_(S)_
(5)
4[Ti] + 2[C] + 2[S] = Ti_4_C_2_S_2(S)_(6)

Finally, as the above reactions proceed, part of the large-size inclusions float to the top. The other Ce inclusions generated during Stage II act as heterogeneous nucleus cores. This result suggests that RE elements can modify inclusion morphology and purify molten steel, further increasing casting slab quality control, as shown in Figure 13 (Stage III).

Based on the reasoning above, combined with Figure 2, the central segregation morphology of the casting slab almost disappears after the addition of RE Ce. According to in-situ statistical analysis, it was found that the addition of RE elements reduces the segregation of easily segregated elements such as C, S, and P, significantly weakening the central segregation defect [23,46]. This study also shows that the formed CeAlO_3_ and Ce_2_O_2_S inclusions have a low degree of mismatch with the Fe matrix, increasing the effectiveness of the nucleation core and improving the solidification structure of C-Mn cryogenic vessel steels. Moreover, due to the strong surface activity of the Ce element, it is segregated in the austenite grain boundary, reducing the interface energy, weakening the driving force for nucleation, refining the structure, and improving the quality of the casting slab. After the addition of composite La and Ce, many high melting point (Ce, La)_2_O_2_S secondary-particles can be used as an effective non-uniform nucleation core, which increases the proportion of equiaxed crystals in the solidified structure, reduces the size of the equiaxed crystals, and significantly refines the solidified structure of the alloy [47].

When RE atoms are dissolved in the matrix, a damping peak of solute atoms segregating at the grain boundary appears in the high-temperature zone. When the RE content exceeds the maximum solid solution of the matrix, it will react with other alloying elements, forming a compound [48]. Therefore, it can be concluded that the addition of 0.0009 wt.% Ce reduces central segregation, accelerates the formation of needle-shaped ferrite, increases the proportion of pearlite, and modifies Al_2_O_3_ and MnS+Ti_4_C_2_S_2_ inclusion morphology in the C-Mn cryogenic vessel steels.

## 4. Conclusions

In industrial production, when the RE amount is over 0.0015 wt.%, the nozzle of the continuous casting mold will appear as a nodule, which is not conducive to the efficient production of long casting times. In this study, through comparative experimental studies, it was found that when the Ce amount is 0.0009 wt.%, the central segregation, microstructure characteristics, and inclusion control of C-Mn cryogenic vessel steel benefited, which can further provide theoretical guidance for the industrial production of RE steel. Based on the above analysis, the main results can be summarized as follows:

(1) During the solidification of the RE-treated steels, the central segregation was significantly improved after adding Ce, reaching the lowest level when the content of Ce was 0.0009 wt.%.

(2) Comparing the microstructures of the casting slabs at different positions in the thickness direction after the addition of Ce, the microstructure and inclusions from the casting slab to the hot-rolled steel strip had a certain heritability, the ferrite became needle-shaped, and the proportion of pearlite increased.

(3) Experimental observation and thermodynamic calculations may indicate that Ce can modify the sharp-angled Al_2_O_3_ inclusions into ellipsoid CeAlO_3_ composite inclusions, and that and the formation of MnS+Ti_4_C_2_S_2_ inclusions was reduced in favor of forming Ce_2_O_2_S + Ti_4_C_2_S_2_ spherical inclusions. Meanwhile, finer Ce_2_O_2_S inclusions can serve as heterogeneous nucleation points and reduce the genetic harm of inclusions.

## Figures and Tables

**Figure 1 materials-14-05262-f001:**
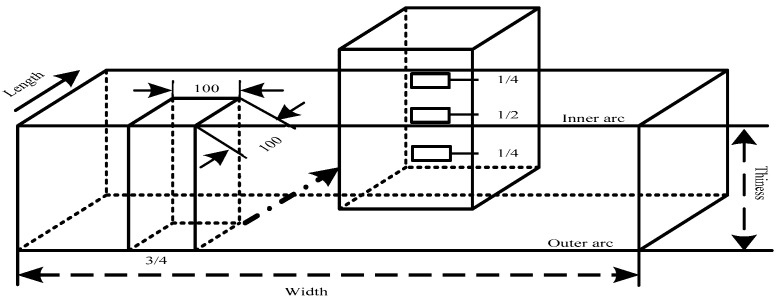
Casting slab sampling schematic diagram.

**Figure 2 materials-14-05262-f002:**
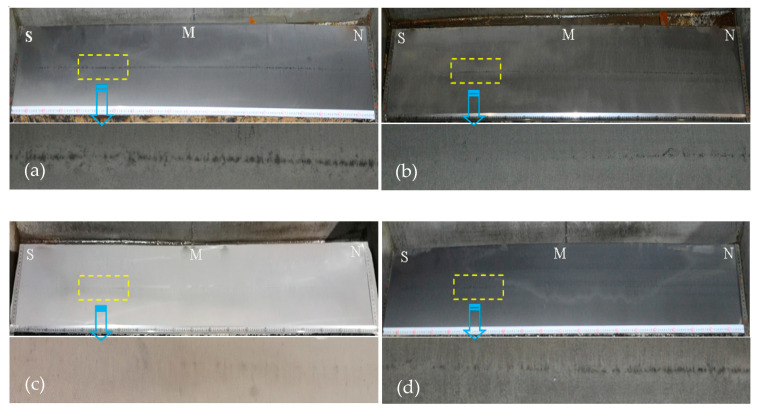
Casting slab quality results, (**a**) 0 wt.% Ce; (**b**) 0. 0006 wt.% RE; (**c**) 0.0009 wt.% Ce; and (**d**) 0.0013 wt.% Ce.

**Figure 3 materials-14-05262-f003:**
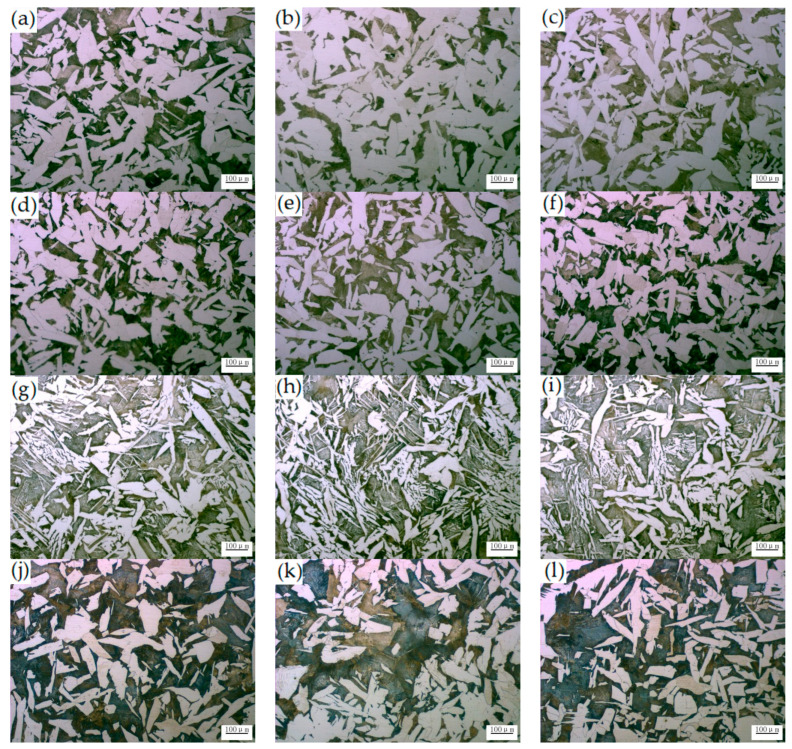
No Ce and different Ce microstructures of the casting slab in the direction of thickness. (**a**–**c**) 0 wt.% Ce, respectively represents 1/4 inner arc position, casting slab center, and outer arc 1/4; (**d**–**f**) 0.0006 wt.% Ce, respectively represents 1/4 inner arc position, casting slab center, and outer arc 1/4; (**g**–**i**) 0.0009 wt.% Ce, respectively represents 1/4 inner arc position, casting slab center, and outer arc 1/4; (**j**–**l**) 0.0013 wt.% Ce, respectively represents 1/4 inner arc position, casting slab center, and outer arc 1/4.

**Figure 4 materials-14-05262-f004:**
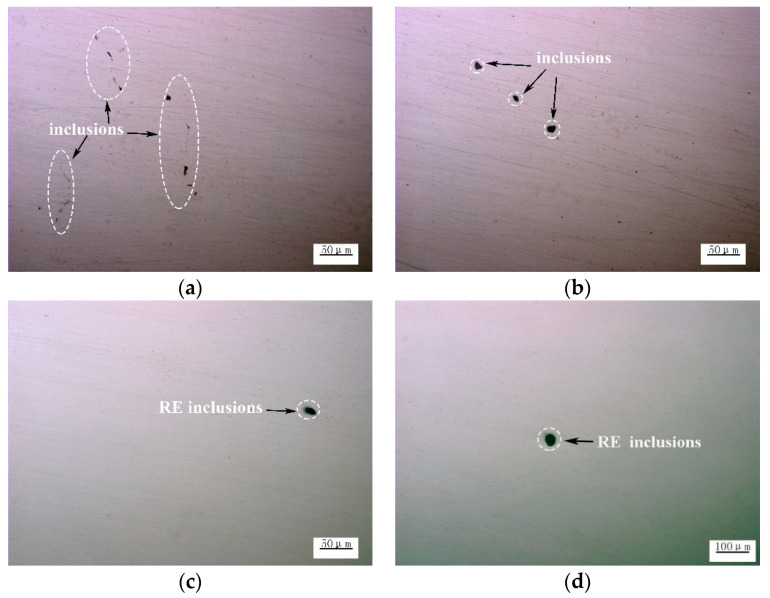
No Ce and different Ce typical inclusions morphology of the casting slab; (**a**) and (**b**) represent 0 wt.% Ce; (**c**) represents 0.0006 wt.% Ce; (**d**) represents 0.0009 wt.% Ce; and (**e**) represents 0.0013 wt.% Ce.

**Figure 5 materials-14-05262-f005:**
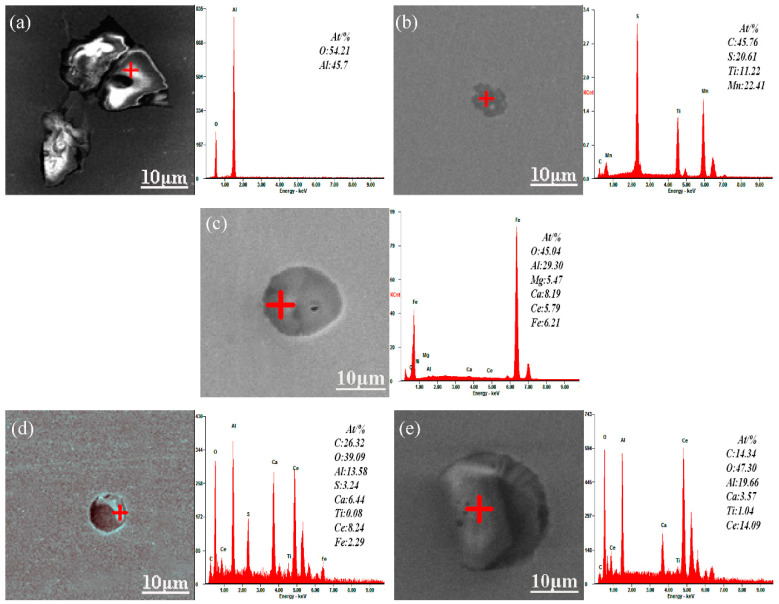
Inclusion morphology and spectrum results of the casting slab; (**a**) and (**b**) represent 0 wt.% Ce; (**c**) represents 0.0006 wt.% Ce; (**d**) represents 0.0009 wt.% Ce; and (**e**) represents 0.0013 wt.% Ce.

**Figure 6 materials-14-05262-f006:**
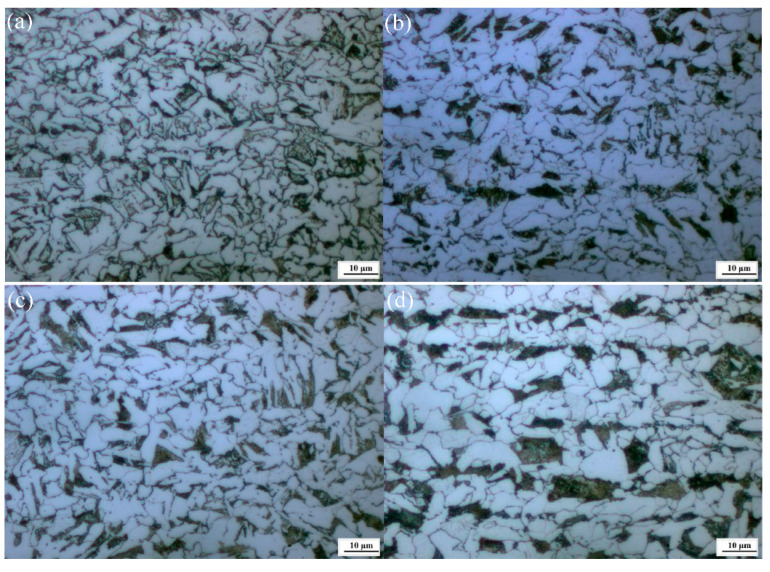
Microstructures of hot rolled steel strips; (**a**) 0 wt.% Ce; (**b**) 0. 0006 wt.% Ce; (**c**) 0.0009 wt.% Ce; (**d**) 0.0013 wt.% Ce.

**Figure 7 materials-14-05262-f007:**
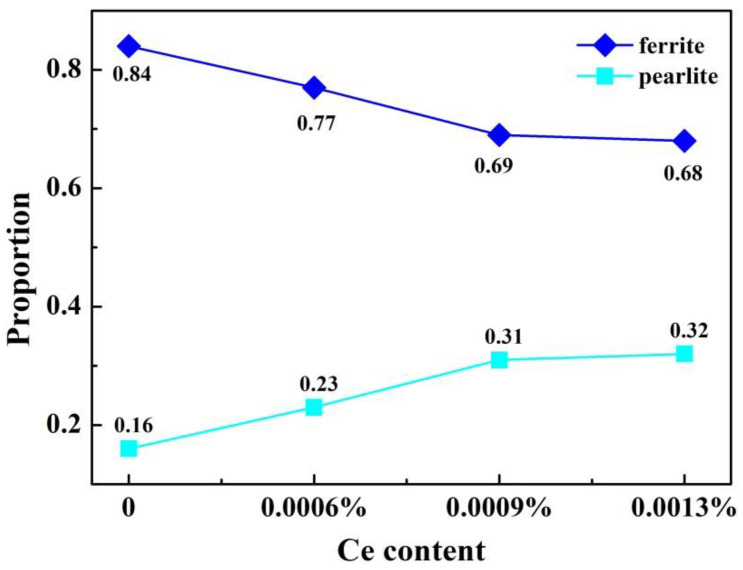
The statistics of ferrite and pearlite in various samples, with and without Ce content.

**Figure 8 materials-14-05262-f008:**
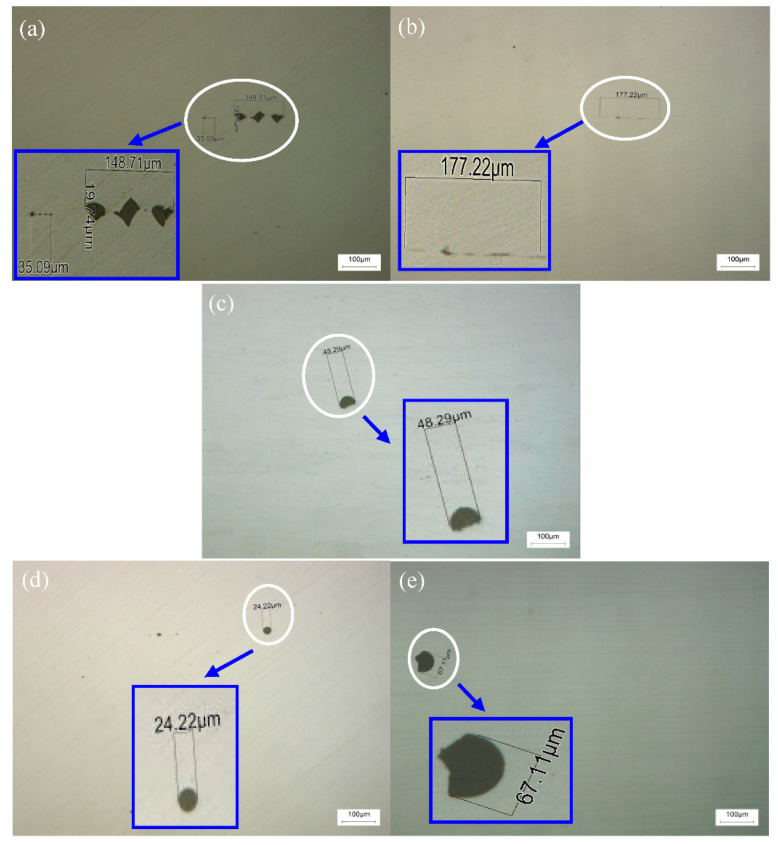
Non-metallic inclusion morphorgy of hot rolled steel strips; (**a**) and (**b**) represent 0 wt.% Ce; (**c**) represents 0.0006 wt.% Ce; (**d**) represents 0.0009 wt.% Ce; and (**e**) represents 0.0013 wt.% Ce.

**Figure 9 materials-14-05262-f009:**
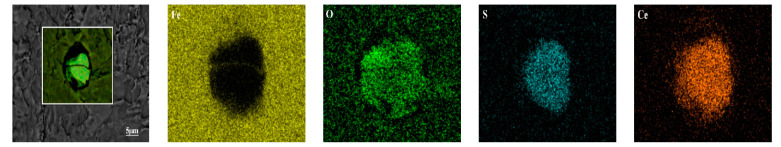
EDS results of the typical inclusions observed in the sample with 0.0009 wt.% Ce.

**Figure 10 materials-14-05262-f010:**
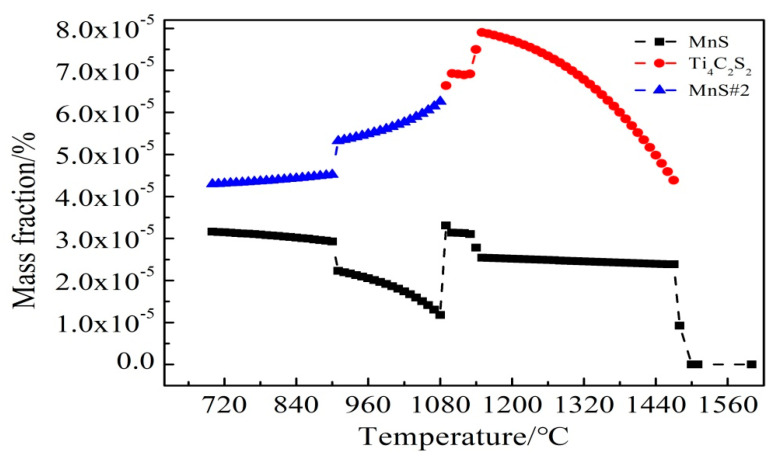
No Ce formation of sulfide.

**Figure 11 materials-14-05262-f011:**
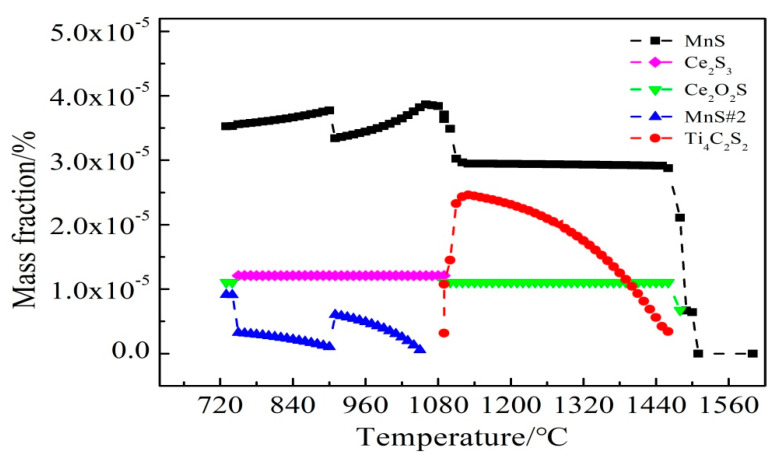
Various Ce formation of sulfide.

**Figure 12 materials-14-05262-f012:**
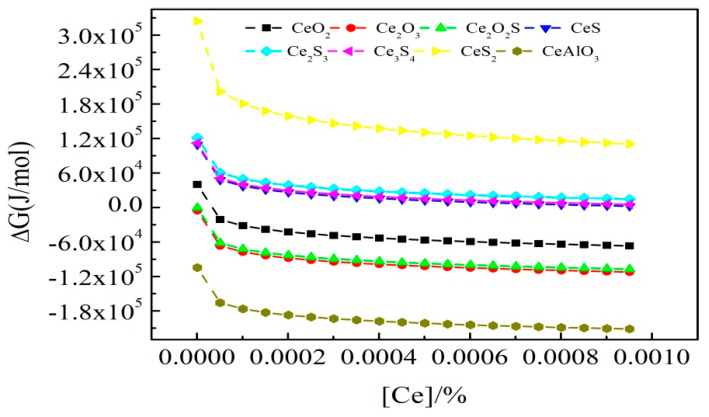
Relation between reaction Gibbs free energy and Ce content.

**Figure 13 materials-14-05262-f013:**
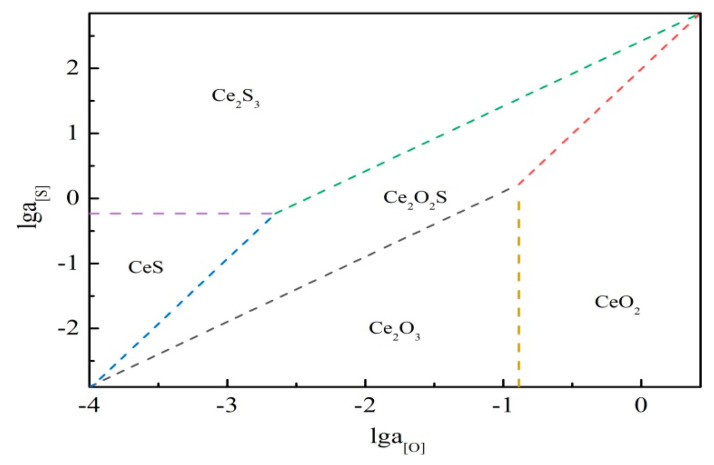
Ce-S-O inclusions generation diagram.

**Figure 14 materials-14-05262-f014:**
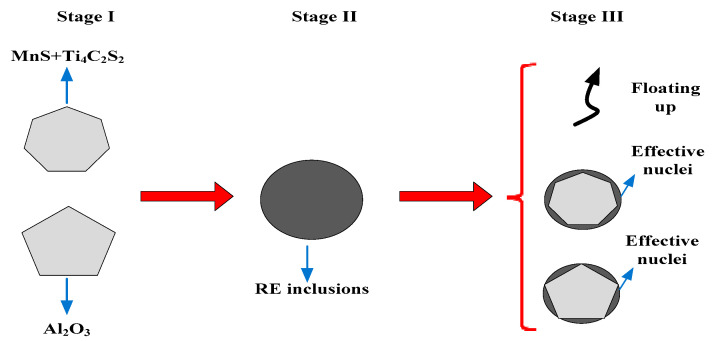
Schematic diagram of the evolution process of inclusions with RE-Ce alloy treatment.

**Table 1 materials-14-05262-t001:** Chemical composition (wt.%).

No.	C	Si	Mn	P	S	Ti	Als ^1^	Ca	Ce
I	0.18	0.16	1.45	0.010	0.003	0.017	0.040	0.0020	0
II	0.18	0.15	1.48	0.010	0.002	0.018	0.039	0.0020	0.0006
III	0.18	0.15	1.48	0.010	0.002	0.018	0.039	0.0020	0.0009
IV	0.18	0.15	1.48	0.010	0.002	0.018	0.039	0.0020	0.0013

^1^: Als, acid-soluble aluminum.

**Table 2 materials-14-05262-t002:** The low magnification judgment results.

Content	Central Segregation	Center Porosity
0 Ce	B 1.0	1.0
0.0006 wt.% Ce	B 1.0	1.0
0.0009 wt.% Ce	C 1.0	1.0
0.0013 wt.% Ce	B 1.0	1.0

**Table 3 materials-14-05262-t003:** Inclusion conversion relationship and formation conditions.

Reactions	Equilibrium Constant	The Relation between a_[O]_ and a_[S]_
2CeO_2_ = Ce_2_O_3_ + [O]	0.12866	a_[O]_ = 0.12866
Ce_2_O_3_ + [S] = Ce_2_O_2_S + [O]	0.079	a_[S]_ = 12.67 × a_[O]_
2CeO_2_ + [S] = Ce_2_O_2_S + 2[O]	0.01017	a_[S]_ = 98.3284 × a_[O]_^2^
Ce_2_O_2_S + [S] = 2CeS + 2[O]	0.8505 × 10^−5^	a_[S]_ = 1.1758 × 10^5^ × a_[O]_^2^
Ce_2_O_2_S + 2[S] = Ce_2_S_3_ + 2[O]	1.452 × 10^−5^	a_[S]_ = 262.433a_[O]_
Ce_2_S_3_ = 2CeS + [S]	0.5857	a_[S]_ = 0.5857

## Data Availability

Data sharing not applicable. No new data were created or analyzed in this study. Data sharing is not applicable to this article.

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
