# Peer review of "Effect of Cerium on the Microstructure and Inclusion Evolution of C-Mn Cryogenic Vessel Steels"

_materials, 2021, doi:10.3390/ma14185262_

Round 1

Reviewer 1 Report

The authors should consider suggestions and answer the questions:

The introduction should be expanded. When authors write:

„The majority of the publication found that different rare earth (RE) elements purify molten steel, modify inclusion morphology, increase cryogenic toughness and improve corrosion resistance in different steel series[6-10].”

It supposes to be said what kind of different RE they referrs to and what effect was obtained in referred studies.

What is the „ferrority alloy”? Did you have on mind the ferroalloy? Please explain and also add the complete chemical composition of this RE alloy mixture.

Why did you write  „… and the content of rare earth in the steel was detected by chemical analysis…” if there are no other RE elements in the alloy than Ce according to table 1.

Figure 1 has horrible quality. The captions are illegible. Also, the sample cut presented on the left suggests the trapezium shape of the sample. I think it may be some mistake in the drawing. It has to be corrected and changed.

Photographs in figure 2 do not show the macrostructure of the casting. The structure is hardly visible. Those are only some pictures of samples.

Explain the sentence:

„Table 2 quantitatively shows the judgment results of the casting slab.”

It is very unclear what the authors had in mind.

The caption that was included under table 2 should not be there. It supposes to be in the text before the table. Also, it is a very strange kind of analysis. I am not sure if this is adequate.

The sentence „With the increase of RE Ce addition, the amount of the pearlite increase and the ferrite finer.” has to be rewritten. I am guessing that authors had in mind a decrease in the amount of ferrite or refinement of ferrite... Anyway, the sentence is unclear and improperly written.

In the sentence „Specific inclusion morphologies are shown in Figure6(c),6(d), and 6(e), the Al2O3 or MnS inclusions wrapped by Ce inclusions were observed.” I think it should be 5(c), 5(d), etc…

Also, there is no Mn in chemical composition presented in Figure c,d,e, so why do you even suggest it may be MnS. Moreover, it can not be told after only the EDS analysis. There are a lot of other elements in presented inclusions (Ti, Ca, C…). You can not write that those are Al2O3 or MnS.

According to the EDS chemical analysis in figure 5, the inclusions presented in Figures 4 a and b and figures 5 a and b may be some different inclusions that can also be present in the samples presented in figures c, d, e. The differences in chemical compositions in those are huge.

For analysis from figures 3 and 6, some quantitative analysis should be done to prove the point of changes in the amount of different phases and their morphology.

In figure 7a the numbers are hardly visible. Also, I can not see the inclusion in 7b. Where is it? It looks like a scratch on the sample.

In Figure 8 caption is wrong. It is the EDS map analysis of the inclusion which contains Ce, O, and S. Also, how does it correspond with the analysis from figure 5 that looks kind of different?

All manuscript is kind of chaotic and it needs a lot of improvement. After the work that was presented, it can be only concluded that Ce addition may cause the creation of spherical Ce containing phases that may influence refinement of the microstructure. And that is all, nothing more.

Reviewer 2 Report

This article details work on the effect of cerium on the microstructure and inclusion evolution of C-Mn cryogenic vessel steels. Their results indicate that the presence of Ce reduces the size of ferrite and improves the pearlite morphology. Interesting work, but some issues need to be resolved.

1. There should be more discussion about the effects of impurities in steels in the introduction section. For reference, please see the following examples:

Metall. Mater. Trans. B-Proc. Metall. Mater. Proc. Sci.   34, Materials Science and Engineering: A 753 (2019) 135-145

2. Some figures need editing. An example is in Figure 5, where it appears that green, wavy lines corresponding to typos is in the figure legends. Also, please refrain from using the color light yellow in the graph as it is very hard to see.

3. I did not see any discussion of Thermo-calc software in the experimental section. This is a very important detail that should be mentioned there. 

4. What does "central looseness" mean, as shown in Table 2? There should be a brief explanation on this concept.

5. References are needed for your discussion on how the addition of Ce can change the casting slab's structural characteristics.

6. There are some moderate errors in the English throughout the text. Please fix this.

Round 2

Reviewer 1 Report

The article still has some flaws, but the work is improved and corrected.

-The Chinese Criterion GB/T 24178-2009 should be mentioned in the manuscript. It might be not very common for readers.

-Rewrite the sentence „Specific inclusion morphologies are shown in Figure 5(c), 5(d) and 5(e), Al2O3 or MnS+Ti-C-S angular inclusions were not found, but the main types of inclusions with Ce addition were Al-O-Ce-Mg-Ca, Al-O-Ce-C-Ca-S, and Al-O-Ce-C-Ca.” It is unclear and needs some English corrections.

-Include in the manuscript what kind of image analysis software was used to determine the proportion between ferrite and pearlite.

-The whole text needs a professional English proofreading. Some sentences are long and make no sense. Spellcheck is also required.

Reviewer 2 Report

The authors appear to have properly addressed my comments.

Author Response

This manuscript is a resubmission of an earlier submission. The following is a list of the peer review reports and author responses from that submission.

Round 1

Reviewer 1 Report

This manuscript discusses the effects of Ce on the casting slab quality, microstructure, and inclusion evolution mechanisms of a cryogenic vessel steel. It was reported that Ce promoted the austenite to pearlite phase transformation. Unfortunately, the quality of the manuscript is seriously lacking, see comments below.

  1. The quality of graphs needs serious improvements as some of the text was unreadable.
  2. Also, what are the units on both images of Figure 2?
  3. The experimental section is lacking serious details An example includes not spelling out some acronyms such as RH and KR.
  4. Also, much more description is needed regarding the scanning electron microscopy and energy dispersive spectrometer work such as what instruments were used.
  5. Several grammatical errors throughout the manuscript. For instance, the second paragraph of Section 3.2 consists of only one very long sentence. Paragraphs should be at least three sentences long as a rule of thumb.
  6. A detailed discussion of results is lacking from the results and discussion section.
  7. Other information is missing such as references to the Thermo-Calc thermodynamics software.
  8. The authors should reduce the amount of y-axis values in Figure 8 to make the graph more clear for the reader.
  9. Equations are misaligned in the text, such as on Page 9.
  10. Please provide references for how they obtained the corresponding chemical equations.

Reviewer 2 Report

The article needs a lot of corrections. Replenishment of the content and experiment improvement is required.  

Abstract

In Sentence

“With the addition of cerium, RE can modify Al2O3 and MnS+Ti4C2S2 inclusions into ellipsoid…” it is not clear if authors talk about only cerium addition or all RE elements. This sentence should be revised and write again.

In the same sentence it is written:

“…RE inclusions deform more easily during hot rolling, reducing the harm of the inclusions.” Did authors mean RE inclusions or phases with Ce which are mentioned before:

“…ellipsoid CeAlO3 and spherical CeAlO3+Ce2O2S+Ti4C2S2…”.

Those issues have to be clarified.

Introduction

The introduction should be expanded and improved.

Authors still use “RE”, but only cerium was the analyzed addition. When authors write:

“Previous researchers have proposed that RE influences the casting slab’s microstructure. For example, with the addition of RE, the equiaxed crystal ratio increased, the columnar crystals were suppressed and refined, and the material’s comprehensive properties improved.[14-16]” it should be clarified which kind of RE elements are analyzed by the authors of referred work. Also, a dot should be after bracket “[14-16].”

The other example of the same issue is present in:

“The majority of released reports found that rare earth (RE) elements purify molten steel, modify inclusion morphology, increase cryogenic toughness and improve corrosion resistance in different steel series[6-10].” References  7 and 9 talk only about lanthanum addition, not cerium. It should be mentioned.

Experimental

In sentence:

“The RE Ce alloys were added into the ladle…” explain RE Ce alloys. You should add the table with the chemical composition of the added mixture.

To table 1 wrote how the chemical composition was measured, on what kind of device, etc.

Also, all values for each element of composition in table 1 should be presented with the same accuracy, if the same device was used for measurement.

Moreover, was the % value of Ce measured on the casting, or was it the wt% calculated for the wight of charge as was mentioned in the abstract? Because it will not be the same after casting. What is “Als”?

In Figure 1 the presentation is unclear. Captions on the sample are too small when on the other hand captions under the scheme are too big. Also, it seems like it is drawn wrong. If the sample was cut the way that is presented it would be in the shape of a trapezium. It has to be corrected. Also, Figure 1 b is incomprehensible, even with the explanation in the text. This should be improved.

The sentence

“The sample were mechanically ground (from #400~#1000) and then polished, using 4% nital acid alcohol corrosion solution.” should be rewritten. It should be written that sample was etched using 4% nital…

In:

“Optical microscope (OM) analysis was used to observe the microstructures. The inclusion distribution and morphology were determined using a scanning electron microscope (SEM) and an energy dispersive spectrometer (EDS).” You have to write, what type of devices you used, from what company, etc. according to the MDPI standards.

Results and Discussion

3.1. Effect of Ce on The Casting Slab Quality

In:

“RE elements have been widely used in the steelmaking process to improve steel quality[17].” Explain how it improves quality based on reference. Also, this part should be in the introduction.

In Figure 2 captions under photographs are too big, the photographs have poor quality, they are too dark and the visibility is low. Should be improved. Also, there is no macrostructure visible in those pictures, so the caption is inadequate.

The explanation is unclear or there is no explanation to table 2. What does it present? What results? What is “C”, “B”? The Description above the table is insufficient and incomprehensible. The caption does not say anything.

Part:

“Thus increasing the heterogeneous nuclei particles[18-20], improving the solidification structure, and reducing the center segregation of the casting slab.” must be developed.

Results and Discussion

3.2. Effect of Ce on Microstructure

In:

“Figure 3 shows the microstructure without RE at different positions of the casting slab thickness, and Figure 4 shows the microstructure after the addition 0.0009wt.% Ce at varying positions within the casting slab thickness.” Explain “varying positions”.

Explain “RE composite inclusion.”

acicular ferrite not “aciculate ferrite”

In Figures 3 and 4 scale is too small and hardly visible. Also rewrite captions correctly, there are some errors in there: “…(a) 1/4 casting thickness of the inner arc inner arc;…”

Results and Discussion

3.3. Effect of Ce on Inclusion Evolution

In Figure 5. Photographs have very low quality, the magnification is too small. There is no morphology of phases visible. Scale is also hardly visible.

The sentence:

“Moreover, it was found that the fine inclusions after the addition of Ce and the samples without Ce had the largest size.” is unclear, should be revised.

In

Figure 6 EDS spectrum is too small and illegible. Moreover, it can not be determined after EDS analysis that those are exactly those compounds. The authors are too sure about the results only from EDS. O, S, C are light elements. It is hard to analyze them only by EDS on SEM. Besides, “Al2O3” presented in figure 6 looks the same as CeAlO3 presented below. Moreover, it has the same light reflex characteristic for compounds with RE elements like Ce. It would be good to ask what elements were excluded from the EDS analysis at this point? Those results look strange.

Results and Discussion

3.4. Ce Inclusion Formed Thermodynamic Mechanism

According to the authors notice the Thermo-Calc analysis does not exactly correspond with what they conclude after EDS analysis. It makes conclusions after EDS even more unreliable.

Results and Discussion

3.5. Inclusion Formation Mechanisms with the Addition of Ce

This part of the discussion is the most valuable part included in the manuscript.

In Equations 1-6 font size should be the same as in the text of the manuscript.

Conclusions

Unfortunately, the conclusions are not supported by research results.